# Dataset Protection via Watermarked Canaries in Retrieval-Augmented LLMs

## Abstract

Retrieval-Augmented Generation (RAG) has become an effective method for enhancing large language models (LLMs) with up-to-date knowledge. However, it may pose a significant risk of copyright infringement, as IP datasets may be incorporated into the knowledge database by malicious Retrieval-Augmented LLMs (RA-LLMs) without authorization. To protect the rights of the dataset owner, an effective dataset membership inference algorithm for RA-LLMs is needed. In this work, we introduce a novel approach, *CanaryTrace*, to safeguard the ownership of text datasets and effectively detect unauthorized use by the RA-LLMs. Our approach preserves the original data completely unchanged while protecting it by inserting specifically designed canary documents into the IP dataset. These canary documents are created with synthetic content and embedded watermarks to ensure uniqueness, consistency, and statistical provability. During the detection process, unauthorized usage is identified by querying the canary documents and analyzing the responses of RA-LLMs for statistical evidence of the embedded watermark. Our experimental results demonstrate high query efficiency, detectability, and consistency, along with minimal perturbation to the original dataset, all without compromising the performance of the RAG system.

## 1 Introduction

Retrieval-Augmented Generation (RAG) enables large language models (LLMs) to dynamically retrieve and integrate external knowledge, extending their capabilities beyond static training data to address up-to-date and domain-specific tasks. However, the reliance on external datasets in RAG raises potential concerns about dataset security and intellectual property (IP) rights. Unauthorized use or replication of proprietary datasets can lead to IP infringement and misuse, posing significant risks for data owners. This erosion of IP protection could, in turn, negatively affect economic efficiency (Ma, 2022). Therefore, safeguarding datasets in RAG systems is a critical challenge requiring methods that ensure ethical usage while preserving the dataset's original utility. One effective protection strategy is to perform pre-release operations on the dataset that enable the data owner to efficiently identify unauthorized use while maintaining the utility of the original IP dataset for authorized RAG systems.

Specifically, embedding a watermark into the IP dataset and detecting it through the outputs of the trained LLMs offers an effective solution to address this issue. Wei et al. (2024) proposes inserting random character sequences into the IP dataset, enabling watermark detection by evaluating the loss of LLMs on these random character sequences. However, this approach embeds a visible watermark into the IP dataset, making it susceptible to being easily identified and removed (Liu et al., 2024c). Furthermore, the detection process relies on accessing the logits of LLMs, which are often inaccessible, e.g., GPT-4 (Brown et al., 2020). To make the watermark more imperceptible, Jovanović et al. (2024) embeds an invisible watermark by paraphrasing each document in the IP dataset. However, the paraphrasing process may introduce significant distortions by altering the original dataset. Moreover, the capacity for watermark embedding varies across IP datasets, and *low-entropy* datasets often lack sufficient redundancy, making it hard to embed watermarks via paraphrasing without compromising their meaning or functionality.

These works highlight the importance of conducting imperceptible pre-release operations on IP datasets to enable effective detection. However, such operations must not compromise the original

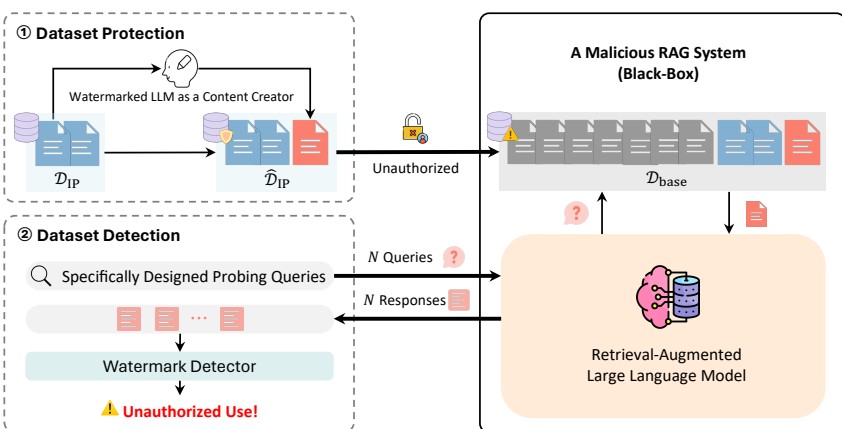

Figure 1: Overview of our *CanaryTrace*. In the dataset protection stage, we generate watermarked and synthetic canary documents based on the attributes of the documents from the IP dataset to form a protected dataset. A malicious RAG system may integrate an IP dataset into its base dataset without obtaining permission from the data owner. During the detection stage, the data owner can conduct black-box queries (without logits information) targeting these canary documents using specifically crafted questions and analyze the model responses to detect the presence of the watermark.

utility of the IP dataset in RAG. The most straightforward way to preserve utility is to keep the original data completely unchanged. Given that existing methods either compromise imperceptibility or significantly alter the original data, we ask: *Can we embed a imperceptible and effective watermark to reliably detect unauthorized usage while keeping the original data completely unchanged?*

In this paper, we formulate the problem as Dataset Membership Inference for Retrieval-Augmented Generation of LLMs (DMI-RAG) and propose an effective solution. Our core idea is to embed a few watermarked *canaries*[1] into the IP dataset, enabling the dataset owner to verify their presence in suspected RA-LLMs. If the watermark embedded in the canaries is detected in the responses of RA-LLMs, the data owner can attribute their presence to unauthorized usage. This operation offers two advantages: 1) maintaining accuracy and nuance: by preserving the original IP dataset, its quality and integrity are maintained, ensuring that the precise wording and stylistic elements remain intact; 2) ensuring the reliable detection for low-entropy dataset: by carefully designing synthetic canaries, we can embed watermarks without altering the original dataset, ensuring effective detection.

Specifically, these canaries are designed to be inconspicuous enough to evade detection or removal by malicious RA-LLMs without impacting the dataset's functionality or performance, while remaining distinctive enough to act as reliable markers for ownership verification. To achieve these properties, we use a watermarked LLM (Zhao et al., 2023) as a content creator to synthesize fictional canary documents that align with the attributes of the IP dataset. This attribute-based synthesis ensures consistency between the canary documents and the IP dataset. Their fictional nature ensures uniqueness, minimizing semantic overlap with other documents in the RAG dataset and improving the accuracy of canary document retrieval. The watermarked LLM embeds an invisible watermark into the canary documents, allowing the watermark to diffuse into the model responses when the canary documents are retrieved. Utilizing the robustness of the watermarking, the data owner can use the watermark in the responses as statistical evidence to detect unauthorized use of the protected dataset. In summary, our main contributions are listed as follows:

- We propose a dataset protection framework in a black-box setting that preserves the IP dataset's documents entirely intact while achieving high detection performance through LLM watermarking embedded in carefully designed canaries.

- We propose an automatic canary synthesis method that leverages the attributes of data from the IP dataset, ensuring high consistency with the original dataset.

- We conduct extensive experiments on various datasets and compare our method with baseline approaches. The results demonstrate that (1) our method achieves a 100% query accuracy for retrieving the canary documents on the NFCorpus dataset (Figure 3). (2) We achieve a 100% TPR@1%FPR with only 12 queries to the suspicious RAG system (Table 1). (3) Our method

---

[1]something acting as informers or decoys for the data owner.

preserves the performance of downstream RAG tasks (Table 4). (4) We demonstrate that the proposed method maintains its effectiveness under scenarios where only a small subset of the protected dataset is stolen (Appendix A.1).

## 2 BACKGROUND AND RELATED WORK

**Retrieval Augmented Generation.** RAG (Karpukhin et al., 2020; Xiong et al., 2020b; Lewis et al., 2020) integrates information retrieval with natural language generation to enhance the response quality and relevance (Gao et al., 2023). In general, a RAG system involves three key components: the retriever $\mathcal{R}$, the generator $\mathcal{M}_0$, and a knowledge database $\mathcal{D}_{\text{base}}$. Given a query $q$, the retriever maps the input and documents in the knowledge database into embeddings within the same space. It then searches the knowledge database to retrieve $K$ most relevant documents, $\{d_1, ... d_K\} = \mathcal{R}(q, \mathcal{D}_{\text{base}})$, based on the distance metric like cosine-similarity. In the generation phase, the generator produces the response given the query and the retrieved documents, i.e., $y = \mathcal{M}_0(d_1, ..., d_K, q)$.

**LLMs Watermarking.** LLM watermarking embeds watermark into the text throughout the entire generation process (Li et al., 2024b; He et al., 2024; Li et al., 2024a; Zhao et al., 2024a;b; Fu et al., 2024; Giboulot & Furon, 2024; Wu et al., 2023; He et al., 2025; Wu et al., 2024; Chen et al., 2025; Chakraborty et al., 2024; Chang et al., 2024a; Liu et al.; 2024a; Pan et al., 2024a;b; Liu et al., 2023a; Zhu et al., 2024). This is typically achieved by either perturbing the logits of the LLM (Liu & Bu, 2024; Kirchenbauer et al., 2023; Zhao et al., 2023) or manipulating the sampling process (Kuditipudi et al., 2023; Christ et al., 2024). In particular, robustness is a critical property of LLM watermarking, enabling the detection of watermarks even after significant text modifications. For the DMI-RAG task, a robust watermark is essential to ensure its persistence from the watermarked document to the response of the RAG system. Therefore, among all existing LLM watermarking methods, we adopt the watermarking by (Zhao et al., 2023) due to its simplicity and provable robustness.

**Data Membership Inference.** Data membership inference (Chang et al., 2024b) for RAG (Liu et al., 2024b) aims to determine whether a specific instance is included in the knowledge dataset of RA-LLMs. Li et al. (2024c) introduces a gray-box approach that computes a membership score by combining text–data similarity with generation perplexity. Anderson et al. (2024) propose a method that uses a specially designed prompt to black-box ask whether a dataset member appears in the context and deduces the membership status based on the model's answer.

**Backdoor Attack.** For dataset membership inference in RAG, the dataset owner can proactively perform specific operations on the original dataset to gather evidence of unauthorized usage. Backdoor attack (Chaudhari et al., 2024; Cheng et al., 2024; Chen et al., 2024) is an approach used to identify unauthorized usage by embedding triggers in the dataset. These triggers cause RA-LLMs to produce specified abnormal responses when queried with certain inputs. Zou et al. (2024) proposes injecting malicious texts into the dataset to manipulate RA-LLMs into generating a predetermined incorrect answer for a specific question. In certain respects, our method shares similarities with a backdoor. However, unlike existing backdoor attacks, which compromise the functionality or performance of the original dataset, our approach is specifically designed to avoid such issues.

**Dataset Membership Inference.** Dataset Membership Inference (Maini et al., 2024; Zhang et al., 2024; Maini et al., 2021; Guo et al., 2023) for protection can be achieved through proactive watermark embedding (Liu et al., 2023b; Panaitescu-Liess et al., 2025). Wei et al. (2024) proposes inserting a random sequence repeatedly into the dataset and then computing the loss of the suspicious LLM on this sampled sequence to determine the dataset's presence. Most recently, Jovanović et al. (2024) proposes to use a watermarked LLM to paraphrase each document in the IP-protected dataset and detect the watermark in the responses of RA-LLM. In this paper, we embed *invisible* watermarks into a small number of carefully designed canaries and insert them into the IP-protected dataset. Our approach preserves the original dataset untouched while achieving high detectability.

## 3 DATASET IP PROTECTION

### 3.1 THREAT MODEL

Our threat model consists of two entities: the data owner, who seeks to protect datasets from unauthorized use, and a malicious RA-LLM, which owns a knowledge dataset and attempts to incorporate the IP dataset into its knowledge base without authorization. In our formulation, the data owner

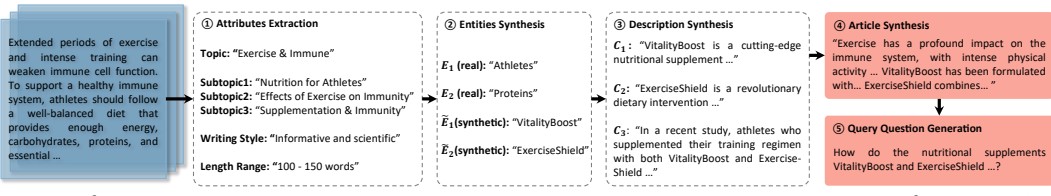

Figure 2: Workflow of our canary dataset synthesis algorithm. The process begins by randomly sampling a document from the IP dataset as a reference. Key attributes are then extracted from the reference document. Using these attributes, descriptions and relationships between synthetic entities are created. Finally, the algorithm outputs the synthetic text and a corresponding query question.

proactively performs operations on the IP dataset before its release, such as embedding invisible watermarks, to enable effective protection. The protected dataset is then accessible to authorized users or may be illegally obtained by unauthorized users. The data owner's capability involves performing black-box dataset membership inference by inputting a limited number of crafted queries to the RA-LLM and analyzing only the generated responses. In the main paper, we assume that unauthorized users steal the entire dataset. However, in Appendix A.1, we also demonstrate that our method remains effective even when only a small subset of the IP dataset is stolen.

## 3.2 PROBLEM FORMULATION

Our high-level idea is to embed watermarked canaries into the protected dataset, serving as evidence of unauthorized use and enabling reliable detection if the dataset is misappropriated. As shown in Figure 1, our method consists of two key components. In the dataset protection phase, the data owner generates watermarked canary documents and inserts them into the IP dataset before the release. In the dataset detection phase, the data owner performs multiple queries to the RA-LLM and provably identifies unauthorized use by framing it as a hypothesis testing problem to distinguish between the following two hypotheses:

$H_0$: The response does not contain watermark; $H_1$: The response contains a watermark.

**Canary Dataset Synthesis.** Specifically, given a RA-LLM $\mathcal{M}_0$ with a knowledge dataset $\mathcal{D}_{\text{base}}$, and an IP dataset $\mathcal{D}_{\text{IP}}$. Our main idea is keeping original $\mathcal{D}_{\text{IP}}$ untouched, and insert a canary dataset $\mathcal{D}_s$ to get a protected dataset $\hat{\mathcal{D}}_{\text{IP}} = \mathcal{D}_{\text{IP}} \cup \mathcal{D}_s$, with the number of documents $|\mathcal{D}_s| \ll |\mathcal{D}_{\text{IP}}|$.

The inserted $\mathcal{D}_s$ should assist the data owner in enhancing query efficiency, providing statistical evidence, and remaining consistent with $\mathcal{D}_{\text{IP}}$. Therefore, it should satisfy three key properties: uniqueness, consistency, and statistical provability. The uniqueness of the canary document minimizes semantic overlap with other documents in $\mathcal{D}_{\text{base}}$ and $\mathcal{D}_{\text{IP}}$, thereby increasing the likelihood that it can be successfully retrieved using specifically designed queries. The consistency of the canary document ensures $\mathcal{D}_s$ shares similar attributes with the original IP dataset, making it more imperceptible. Statistical provability provides statistical evidence of the watermark across multiple queries, therefore enabling decisions with a controlled false alarm.

To achieve these properties, we design a canary data synthesis algorithm to automatically synthesize canary documents $\mathcal{S} : \mathcal{D} \mapsto \mathcal{D}_s \times \mathcal{Q}_s$, which uses a watermarked LLM to synthesize fictional text $d_s \in \mathcal{D}_s$ and the corresponding queries based on the attributes extracted from the sampled data $d \in \mathcal{D}_{\text{IP}}$. This ensures that the synthesized text retains uniqueness and relevance to the original dataset while embedding the necessary watermark for detection. Then, we synthesize query text to form the query dataset $q_s \in \mathcal{Q}_s$, specifically tailored for each $d_s \in \mathcal{D}_s$.

**Dataset Membership Detection.** A watermark detector, $D_w : \mathcal{Y}^* \mapsto \mathbb{R}$, is employed to analyze the outputs $y$ of $\mathcal{M}_0$ generated in response to queries $q_s \in \mathcal{Q}_s$. Specifically, we can determine whether $\mathcal{D}_{\text{IP}} \subseteq \mathcal{D}_{\text{base}}$ if the detector satisfies the condition $D_w(y^{(1)} \oplus y^{(2)} \oplus \cdots \oplus y^{(N)}) \geq \eta$, where $\eta$ is a predefined threshold, $N$ is the number of queries, $\oplus$ represents the concatenation operation.

## 3.3 CANARY DATASET SYNTHESIS

In this section, we present our canary dataset synthesis algorithm $\mathcal{S}$. As shown in Figure 2, it includes: 1) attributes extraction, 2) entity synthesis 3) description synthesis, 4) article synthesis, and 5) query question generation.

**Attributes Extraction.** To ensure consistency with $\mathcal{D}_{\mathsf{IP}}$, each canary synthesis process begins by randomly sampling a document from $d \in \mathcal{D}_{\mathsf{IP}}$. Then, we analyze and extract the key attributes $\mathcal{A} = \{A_1, A_2, ..., A_n\}$ of sampled text using an attribute extraction function, attr_extr_func. This process is facilitated by an LLM $\mathcal{M}$ with appropriately designed prompts: $\mathcal{A} \leftarrow \mathcal{M}(\mathsf{attr\_extr\_func}(d))$.

In our experiments, we extract four attributes from the sampled document $d$: $\{A_1 = \mathsf{Topic}, A_2 = \mathsf{Subtopic}, A_3 = \mathsf{Writing\ Style}, A_4 = \mathsf{Word\ Count}\}$. These attributes ensure that the synthesized documents align with the same topic, writing style, and length as the original dataset.

**Entity Synthesis.** The uniqueness of the canary documents significantly reduces overlap with other documents within $\mathcal{D}_{\mathsf{base}}$ and $\mathcal{D}_{\mathsf{IP}}$, ensuring a clear separation between the embeddings of the canary documents and those of other documents. This distinctiveness greatly enhances the accuracy of retrieving the target canary documents (Figure 3). To ensure the uniqueness of the synthesized text, we extract real entities $\mathcal{E} = \{E_1, ...E_n\}$ from $d$ and create fictional entities $\tilde{\mathcal{E}} = \{\tilde{E}_1, ..., \tilde{E}_n\}$ conditioned on the attributes $\mathcal{A}$ using the LLM with the ent_synth_func: $\{\mathcal{E}, \tilde{\mathcal{E}}\} \leftarrow \mathcal{M}(\mathsf{ent\_synth\_func}(d, \mathcal{A}))$.

**Description Synthesis.** Then, we synthesize the fictional descriptions for each entity in $\tilde{\mathcal{E}}$ and the relations between them $\mathcal{C} = \{C_1, ...C_n\}$ using the LLM with the des_synth_func: $\mathcal{C} \leftarrow \mathcal{M}(\mathsf{des\_synth\_func}(\tilde{\mathcal{E}}, \mathcal{A}))$. This step preserves contextual relevance to $\mathcal{D}_{\mathsf{IP}}$ while ensuring that synthesized text remains distinct and identifiable.

**Article Synthesis.** Finally, we collect the synthesized descriptions $\mathcal{C}$ to generate a new article $d_s$ using the LLM with article_synth_func:

$$d_s \leftarrow \mathcal{M}_w(\mathsf{article\_synth\_func}(\mathcal{C}, \mathcal{A})), \tag{1}$$

where $\mathcal{M}_w$ is a watermarked LLM, which will be detailed in Section 3.4.

**Query Question Generation.** For each $d_s \in \mathcal{D}_s$, we prompt an LLM to generate a question that can only be answered by reading $d_s$ using query_synth_func:

$$q_s \leftarrow \mathcal{M}(\mathsf{query\_synth\_func}(d_s)) \tag{2}$$

The generated $q_s$ is used to retrieve the corresponding $d_s$ from the suspicious RA-LLM during the detection process. All the specific prompts used in our experiments are provided in Appendix C. Examples of the canary documents are provided in Appendix D.

While synthesizing fictional canary documents can markedly improve retrieval accuracy, it also raises the risk of being identified and removed by malicious RA-LLMs that potentially leverage *fact-checking tools*. To alleviate this risk, we introduce a canary pre-filtering method. Specifically, we use a recently released LLM as a powerful fact-checker to screen out more suspicious canaries, ensuring that only those resistant to fact-checking are retained. More details about the pre-filtering method are discussed in Appendix A.2.

## 3.4 WATERMARKING SYNTHETIC DATASET

The process outlined in Section 3.3 introduces the uniqueness and consistency of the synthesized dataset. In this section, we detail the approach for ensuring the provability by integrating a watermarked LLM $\mathcal{M}_w$ into (1) of the algorithm $\mathcal{S}$.

We employ the green-red list watermarking scheme (Zhao et al., 2023), which randomly partitions the LLM's vocabulary $\mathcal{V}$ into the global fixed green ($G$) and red ($R$) lists, such that $|G| = \gamma|\mathcal{V}|$ and $|R| = (1-\gamma)|\mathcal{V}|$, where $\gamma \in (0, 1)$ represents the proportion of tokens assigned to the green list. This scheme slightly increases the logits $\ell$ of tokens in the green list at each time step, such that $\hat{\ell}_t[k] \leftarrow \ell_t[k] + \delta$ for $k \in G$. Therefore, the output of the watermarked LLM will exhibit a higher probability of green tokens. The key assumption here is that the watermark embedded in the document should remain in the output (Sander et al., 2024) of the RA-LLM (which is shown in Section 4) when the watermarked document is retrieved in response to a query. Therefore, it provides a way to determine the dataset membership by analyzing the model's response and calculating the probability of green tokens present in the output.

### 3.5 DATASET MEMBERSHIP INFERENCE

By applying the canary dataset synthesis algorithm $\mathcal{S}$, we can obtain the query dataset $\mathcal{Q}_s$ and canary dataset $\mathcal{D}_s$. The data owner can combine $\mathcal{D}_{\mathsf{IP}}$ and $\mathcal{D}_s$ to construct the new dataset $\hat{\mathcal{D}}_{\mathsf{IP}}$ prior to its release. During the dataset membership inference process, for a suspicious RA-LLM $\mathcal{M}_0$, we first query the model $N$ times using different $q_s^{(i)} \in \mathcal{Q}_s$ and obtain the response $y^{(i)} = \mathcal{M}_0(\mathcal{R}(q_s^{(i)}, \mathcal{D}_{\mathsf{base}}))$. Based on the discussion in Section 3.4, the dataset membership inference problem can be effectively transformed into a watermark detection problem. However, directly detecting the watermark in a single response is challenging because the embedded watermark in $d_s$ becomes significantly weakened after passing through the RAG pipeline. Therefore, we concatenate $N$ responses into a single sequence, denoted as $\mathbf{y} = y^{(1)} \oplus y^{(2)} \oplus \cdots \oplus y^{(N)}$, to enhance the statistical detectability of the watermark in responses.

Specifically, following Kirchenbauer et al. (2023) and Zhao et al. (2023), we detect the watermark by computing the z-statistic of the response $\mathbf{y}$, i.e.,

$$z_{\mathbf{y}} = (|\mathbf{y}|_G - \gamma T)/\sqrt{\gamma(1-\gamma)T}, \tag{3}$$

where $T$ is the number of tokens in $\mathbf{y}$, and $|\mathbf{y}|_G$ is the count of the tokens in green list. The response $\mathbf{y}$ is identified as watermarked if $z_{\mathbf{y}} > \eta$, where $\eta$ is a pre-defined threshold. Therefore, we can equivalently conclude if $\mathcal{D}_{\mathsf{IP}} \subseteq \mathcal{D}$. Moreover, we establish a theoretical guarantee that controls the false-alarm rate, thereby limiting the risk of falsely accusing benign RA-LLMs. More details can be found in Appendix B.

## 4 EXPERIMENTS

### 4.1 EXPERIMENT SETTING

**Implementation Details.** *(1) Model Settings:* For our canary dataset synthesis algorithm, we use GPT4o-mini to extract attributes and generate fictional descriptions. Additionally, we use watermarked Llama-3.1-70B-Instruct to embed the watermark into the synthesized articles. *(2) Watermark Settings:* For the watermarking algorithm, we set $\gamma = 0.5$ and $\delta = 2.0$. *(3) Retriever Settings:* For retriever, we conduct experiments based on the Contriever-ms (Izacard et al., 2021) and ANCE (Xiong et al., 2020a) We select the top $K = 3$ most relevant documents based on the cosine similarity. In our main experiments, each canary document is queried only once with a single question. We investigate the detection performance under the setting where multiple queries are conducted per canary document in Appendix A.4. All experiments are conducted on NVIDIA A100 GPU.

**Baselines.** We compare our methods with two existing dataset membership inference methods leveraging watermark, including Ward (Jovanović et al., 2024), and WWJ (Wei et al., 2024). For Ward, consistent with its default settings, we use KGW (Kirchenbauer et al., 2023) to paraphrase the whole dataset and set $\gamma = 0.5$, $\delta = 3.5$, and window size $h = 2$. We adapt WWJ to our setting; implementation details can be found in Appendix A.3.

**Datasets.** We use the MS MARCO (Nguyen et al., 2016) as the knowledge dataset of RA-LLMs, which is a large-scale real-world web document corpus. In the main results, we set the size of the base dataset to $200,000$. To further demonstrate the effectiveness of our method, we extend this size to 2 million in Appendix A.5. In this setting, we assume that potential semantic overlap may exist between the knowledge dataset and the IP dataset, which often happens in the real world. We evaluate our performance using the bio-medical information retrieval dataset, NFCorpus (Boteva et al., 2016), as the IP dataset. Moreover, we use CQADupStack-Mathematica (Hoogeveen et al., 2015) as IP dataset to evaluate the detection performance of our methods on the low-entropy dataset. We evaluate the downstream performance of different methods on Chinese-poem[2] and DROP dataset (Dua et al., 2019).

**Evaluation Metrics.** To evaluate and compare the performance of different methods, we employ the following evaluation metrics.

---

[2]https://huggingface.co/datasets/xmj2002/tang_poems

Table 1: Detection performance across different methods on NFCorpus dataset with varying query quota. Our method achieves 100% detection performance with only 12 query quota.

| Query Quota | WWJ | | | Ward | | | **Ours** | | |
|---|---|---|---|---|---|---|---|---|---|
| | ROC-AUC | TPR@1%FPR | TPR@10%FPR | ROC-AUC | TPR@1%FPR | TPR@10%FPR | ROC-AUC | TPR@1%FPR | TPR@10%FPR |
| 2 | - | - | - | 0.809 | 0.274 | 0.482 | 0.970 | 0.546 | 0.908 |
| 4 | 0.986 | 0.968 | 0.972 | 0.880 | 0.336 | 0.660 | 0.995 | 0.850 | 0.996 |
| 6 | 0.987 | 0.969 | 0.974 | 0.922 | 0.530 | 0.762 | 0.998 | 0.960 | 1.000 |
| 8 | 0.990 | 0.970 | 0.974 | 0.940 | 0.520 | 0.820 | 0.999 | 0.994 | 1.000 |
| 10 | 0.990 | 0.972 | 0.976 | 0.963 | 0.660 | 0.900 | 0.999 | 0.998 | 1.000 |
| 12 | 0.991 | 0.972 | 0.978 | 0.982 | 0.792 | 0.954 | **1.000** | **1.000** | **1.000** |

- We use Target Retrieval Accuracy to evaluate the proportion of corresponding watermarked documents successfully retrieved by the queries. It is computed as: $\frac{1}{N}\sum_{i=1}^{N} \mathbb{1}(d^{(i)} \in \mathcal{R}(q^{(i)}))$.

- For detectability, the detection performance is assessed using the ROC-AUC value, which measures the ability of a detector to distinguish between classes by evaluating the trade-off between true positive rate (TPR) and false positive rate (FPR) across different thresholds. Moreover, we report detection performance at different FPR values, such as TPR@1%FPR and TPR@10%FPR.

- We evaluate the dataset distortion caused by different dataset protection methods using BLEU (Papineni et al., 2002) and MAUVE (Pillutla et al., 2021) scores.

- The consistency is measured using Perplexity and QuRating (Wettig et al., 2024), which are two methods to curate a dataset. More details can be found in Appendix A.3.

- We calculate the response correctness using GPT4o-mini as a judge to assess the impact of each dataset protection method on the downstream performance of RA-LLMs.

## 4.2 MAIN RESULTS

**Target Retrieval Accuracy.** Target Retrieval Accuracy is a crucial performance metric for the DMI-RAG task. A high Target Retrieval Accuracy ensures that the target watermarked documents can be retrieved with high probability using specifically designed queries. In this experiment, Target Retrieval Accuracy is calculated using 500 different questions to retrieve the corresponding watermarked documents for each method. Specifically, for our method, we insert 500 canary documents into the IP dataset. Figure 3 shows the Target Retrieval Accuracy for the original IP dataset and different baselines. The 'Original' represents the original IP dataset without any modification, where we generate 500 different questions to query different documents in the original IP dataset. As shown in the table, our method outperforms all others,

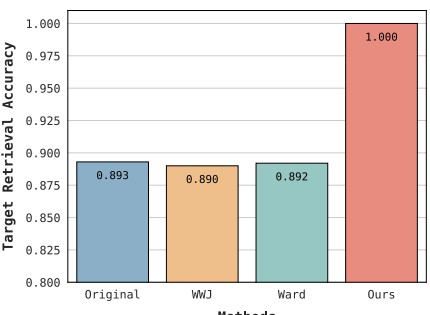

Figure 3: Retrieval Accuracy for different methods. Our method achieves 100% Retrieval Accuracy.

achieving a perfect Retrieval Accuracy of 100%. This is attributed to the insertion of synthetic fictional canaries, which minimizes the potential semantic repetition with other documents in the base dataset. In contrast, the methods employed by WWJ and Ward affect the Target Retrieval Accuracy to varying degrees compared to the original IP dataset.

**Detectability.** We evaluate and compare the detection performance across different query quotas for various methods. Our goal is to achieve superior detection performance with fewer number of queries. As shown in Table 1, our method outperforms the baseline approaches by achieving superior detection performance with a lower query count. In particular, we achieve 100% TPR@1%FPR by only querying the RA-LLM 12 times, while Ward is only 79.2% on the same quota. One main reason is that our method achieves a high Target Retrieval Accuracy.

**Dataset Distortion.** Dataset distortion quantifies the extent of changes introduced by various methods to the original IP dataset. To evaluate this, we use BLEU and MAUVE scores. BLEU assesses the $n$-gram word-level overlap between the original dataset and the protected dataset, while MAUVE evaluates the similarity in distributions between the two datasets. As shown in Table 2, our method achieves the highest BLEU and MAUVE scores compared to other methods, demonstrating minimal distortion to the original dataset.

This is attributed to our approach of keeping the original documents entirely unchanged while ensuring that the inserted canary documents share the same attributes as the original IP dataset. However, Ward modifies the expressions and words in the original dataset, resulting in low BLEU and MAUVE scores. The random sequence inserted by WWJ does not alter the original expressions, resulting in a high BLEU score. However, it introduces a significant distribution shift between the original IP dataset and the protected dataset, leading to a low MAUVE score.

Table 2: Dataset distortion for different methods. Our method achieves the lowest dataset distortion for both BLUE and MAUVE.

| Methods | WWJ | Ward | Ours |
|---|---|---|---|
| BLUE (↑) | 0.981 | 0.132 | **0.997** |
| MAUVE (↑) | 0.004 | 0.340 | **0.999** |

**Consistency.** The consistency of our method is evaluated using two dataset curation methods: perplexity and QuRating. For perplexity, we calculate the perplexity of the canary documents generated by our method, the paraphrased documents produced by Ward, and the documents with inserted random sequences created by WWJ. We set the maximum perplexity value of the original dataset as a threshold to compute the Filtering Rate. In particular, we compute the perplexity using GPT-3. Table 3 shows that our canary documents exhibit a low average perplexity, comparable to that of the original IP dataset, and a 0 Filtering Rate. This demonstrates that our synthetic data is difficult to detect and remove based on the perplexity. However, WWJ shows a high Filtering Rate and average perplexity. This is because the randomly sampled sequences are inconsistent with the natural language patterns expected by a language model, resulting in extremely high perplexity.

Table 3: Perplexity and Filtering Rate across different methods. Our method achieves a perplexity comparable to that of the original dataset and 0% Filtering Rate.

| Methods | Original | WWJ | Ward | Ours |
|---|---|---|---|---|
| Avg Perplexity (↓) | 6.736 | 12.734 | 9.453 | **6.952** |
| Filtering Rate (↓) | - | 0.104 | 0.015 | **0.000** |

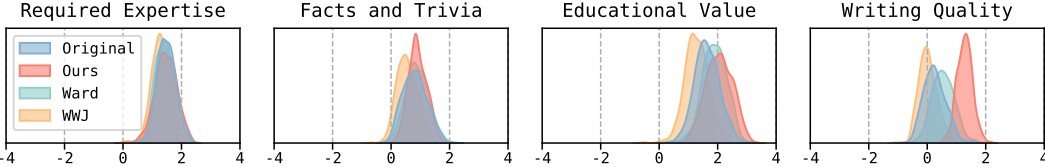

Figure 4: Distribution of quality ratings using QuRating across four aspects for various methods.

For QuRating, we measure the canary documents generated by our method, the paraphrased documents produced by Ward, and the documents with inserted random sequences created by WWJ from four different aspects, obtaining the corresponding scores for each. The dataset distributions across different aspects are shown in Figure 4. As can be seen, our canary documents exhibit higher writing quality, making them less likely to be removed based on this criterion. For the remaining three aspects, our synthetic documents exhibit a similar distribution to the original documents, demonstrating their consistency with the original IP dataset. Both perplexity and QuRating demonstrate the high quality and consistency of our canary documents.

**Downstream Performance.** We further investigate the impact of different methods on the downstream performance by measuring the correctness of the model response. We use two datasets for two different tasks.

For DROP, we evaluate the correctness of the model's responses on the protected dataset, specifically for discrete reasoning and numerical computation tasks. As shown in Table 4, our method maintains the correctness of the original IP dataset. This is because our method preserves the original data without any modifications. In contrast, the correctness of Ward decreases from 0.733 to 0.714. The drop in Ward's performance suggests that paraphrasing may introduce additional errors, negatively affecting downstream tasks.

Table 4: Downstream Performance on two datasets.

| Methods | Dataset | Correctness |
|---|---|---|
| Original | DROP | 0.772 |
| | Chinese-Poem | 0.908 |
| WWJ | DROP | 0.771 |
| | Chinese-Poem | 0.900 |
| Ward | DROP | 0.767 |
| | Chinese-Poem | 0.719 |
| Ours | DROP | 0.772 |
| | Chinese-Poem | 0.908 |

Moreover, for literary datasets, such as poetry, we evaluate correctness in terms of poem appreciation. Specifically, we emphasize the critical importance of preserving the original data intact, as even slight modifications to the words in a poem can significantly alter the author's intended expression

Table 5: Detection performance for different methods on low-entropy dataset (Mathematica) with varying query quota. Our method achieves $100\%$ detection performance with a query quota of just 10, while Ward shows a significantly lower detection performance.

| Query Quota | Ward | | | Ours | | |
|:---:|:---:|:---:|:---:|:---:|:---:|:---:|
| | ROC-AUC | TPR@1%FPR | TPR@10%FPR | ROC-AUC | TPR@1%FPR | TPR@10%FPR |
| 2 | 0.572 | 0.029 | 0.116 | 0.975 | 0.584 | 0.938 |
| 4 | 0.614 | 0.016 | 0.215 | 0.997 | 0.920 | 1.000 |
| 6 | 0.623 | 0.037 | 0.244 | 0.999 | 0.974 | 1.000 |
| 8 | 0.628 | 0.012 | 0.228 | 0.999 | 0.998 | 1.000 |
| 10 | 0.668 | 0.049 | 0.219 | **1.000** | **1.000** | **1.000** |
| 12 | 0.680 | 0.041 | 0.235 | **1.000** | **1.000** | **1.000** |

and emotional tone, thereby impacting the overall interpretation and appreciation of the work. From Table 4, the correctness of Chinese poem appreciation for Ward drops significantly from 0.908 to 0.719, whereas our method maintains the same level of correctness.

### 4.3 DETECTION PERFORMANCE ON HARD CONDITIONS

**Detectability on Low Entropy IP Dataset.** We further evaluate the detection performance of our method on a low-entropy IP dataset, CQADupStack-Mathematica (Hoogeveen et al., 2015), and compare it with Ward. As shown in Table 5, our method maintains a high detection performance, achieving $100\%$ TPR@1%FPR using only 10 query quotas. In contrast, Ward achieves significantly lower detection performance, with $0.049\%$ TPR@1%FPR at the same query quota. This is because low-entropy IP datasets lack redundancy, making it difficult to perform modifications. As a result, paraphrasing can embed only a limited amount of watermark, which fails to persist in the model's responses. In contrast, our method remains unaffected, as we embed watermarks into carefully designed canaries, which are not constrained by the entropy of the original IP dataset.

**Hard System Prompt.** We investigate the detection performance under two types of system prompts: easy and hard. An easy system prompt is straightforward, asking the model to answer the question without imposing additional constraints or complex instructions. In contrast, a hard system prompt includes more restrictive instructions, such as strictly avoiding verbatim text or excessive paraphrasing. Table 1 shows de-

Table 6: Detection performance of our method on hard system prompt.

| Query Quota | ROC-AUC | TPR@1%FPR | TPR@10%FPR |
|:---:|:---:|:---:|:---:|
| 2 | 0.962 | 0.408 | 0.894 |
| 6 | 0.997 | 0.868 | 0.996 |
| 10 | 0.999 | 0.996 | 1.000 |
| 14 | **1.000** | **1.000** | **1.000** |

tection performance under the easy system prompt. In Table 6, we showcase the detection performance under the hard system prompt. The results demonstrate that our method achieves the $100\%$ TPR@1%FPR with 14 query quotas, indicating that the detection performance remains unaffected even with hard system prompt.

**Additional Main Results.** In Appendix A.4, we present two **extra results**: (1) Detection performance when querying a single canary document multiple times. Specifically, in the multiple queries per canary document setting, we show that the data owner can achieve $100\%$ detection performance with as few as 5 canary documents. (2) Robustness of our method under the canary paraphrasing attack and watermarked RA-LLM.

**Ablation Study.** In Appendix A.5, we conduct the following **ablation studies**: (1) The impact of watermark strength on detection performance. (2) Detection performance on different RAG systems. (3) Detection performance with a larger base dataset. (4) Detection performance when $K = 1$. (5) In Appendix A.6, we use a different watermarking approach (Kirchenbauer et al., 2023) to generate the watermarked canaries, further demonstrating the effectiveness of our approach. (6) Since our canary synthesis algorithm is capable of generating highly realistic canaries, we also examine this scenario and analyze its impact on retrieval accuracy.

## 5 CONCLUSION

In this paper, we propose a novel method to protect proprietary datasets from unauthorized use while preserving their original integrity by embedding carefully designed canary documents into the IP-protected dataset. Our experimental results demonstrate the effectiveness of our method in both detectability and consistency, making it a practical and reliable approach for real-world applications.

ETHICS STATEMENT

This work focuses on protecting the IP of text datasets from unauthorized use in RAG systems. Our method does not involve human subjects, personal data, or sensitive information. The proposed canary synthesis process generates fully fictional content, ensuring that no identifiable or private data is introduced. The goal of this research is to promote ethical use of datasets by enabling dataset owners to verify unauthorized usage in a statistically rigorous manner. All experiments are conducted on publicly available benchmark datasets, and we do not release any proprietary or private data.

REPRODUCIBILITY STATEMENT

All algorithms, including the canary synthesis procedure, dataset membership inference method, and statistical testing framework, are described in detail in the main text to ensure reproducibility. Additional implementation details, hyperparameter settings, and ablation studies are provided in the main text and appendix. All datasets used in our experiments are publicly available, and we describe preprocessing steps to facilitate replication.

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

## THE USE OF LLMs

LLMs are used solely as assistive tools to polish the writing and improve the readability of the manuscript. They were not used for research ideation, experimental design, data analysis, or result generation. The authors take full responsibility for the content of this paper.

## A  ADDITIONAL EXPERIMENT RESULTS

### A.1  DISCUSSION ABOUT ONLY A SUBSET OF IP DATASET IS STOLEN

Although we assume the entire IP dataset is stolen in the threat model, our method remains effective even when only a subset of the IP dataset is stolen, as long as a sufficient number of canaries are inserted. Considering a realistic setting where we could query one document multiple times (Appendix A.4), we could realize 100% TPR@1%FPR with 5 canaries. Assuming $|D_{IP}| = 100,000$, the stolen dataset includes at least 5 canaries with 99% probability. We compute the minimum canary set size $|D_s|$ needed for various stolen proportions. As shown in the Table 7, inserting just 0.054% canaries ensures, with 99% probability, that at least 5 will be captured, even if only 20% of the dataset is leaked.

Table 7: Minimum canary set percentage needed for various stolen proportions.

|  | 20% | 40% | 60% | 80% |
|---|---|---|---|---|
| $|D_s|/|D_{IP}|$ | $5.4 \times 10^{-4}$ | $2.5 \times 10^{-4}$ | $1.5 \times 10^{-4}$ | $1.0 \times 10^{-4}$ |

### A.2  CANARY PRE-FILTERING SETTING

To filter out highly suspicious canaries and retain only those resistant to fact-checking, we employed a recently released LLM, Gemini-2.5-Flash-Lite (Comanici et al., 2025), as an automated and powerful fact-checker. It simulates the behavior of malicious RA-LLMs that may leverage fact-checking tools to conduct attacks. Specifically, we synthesized 500 distinct canary documents and asked the model to flag any text containing fictional entities. Of these, 23.8% canary documents escaped detection, demonstrating that nearly one in four canaries remains hidden even under direct scrutiny, which is enough for us to protect the dataset effectively. This pre-filtering step ensures that only high-resilience canaries are retained. The concrete prompt used for canary pre-filtering is presented as follows.

---

**Canary Pre-filtering Prompt**

### Instructions:
You are a fact-checker specialized in spotting whether a text mentions any true, verifiable real-world entities (people, organizations, grant numbers, legal cases, etc.)

When I give you a document, do the following:
1. Scan the entire document for any named entity or specific fact.
2. For each entity, try to check it against reliable public records or your knowledge.
3. Decide **contain_fictional_entity**: output 1 if it contains unverified/fictional entities, 0 if all entities are verifiable/real.
4. Provide a brief **reason** to explain your verdict.
5. Output your answer in JSON format with the following schema:{"contain_fictional _entity": 1/0, "reason": "your reason here"}

### Document:
{*canary*}

---

### A.3 EXTRA EXPERIMENT SETTINGS

**Implementation details of baseline WWJ.** For WWJ, the original method is designed to protect the dataset used for training or fine-tuning. We make several modifications to adapt this method to the DMI-RAG task. Specifically, we randomly sample a watermark sequence $u$ from the ASCII table and then insert the watermark sequence into each document in the IP dataset. Next, similar to our method, we generate questions $(q_u^{(1)}, ..., q_u^{(N)})$ for each document to query the corresponding content. For the detection process, we first use the $q_u^{(i)}$ to compute the loss of the watermark sequence $l_u$. Then, we use the dataset-unrelated questions $(\tilde{q}^{(1)}, ..., \tilde{q}^{(N)})$ to query the RA-LLM to compute the loss of $u$ when retrieved documents do not contain watermark sequence. This process yields the mean $\mu$ and standard deviation $\sigma$ of loss $(\tilde{l}_u^{(1)}, ..., \tilde{l}_u^{(N)})$. The final decision is made by computing the statistic: $z = (l_u - \mu)/\sigma$. However, we note that this baseline requires access to the log probabilities of the suspicious model's predictions, which is not available in the black-box setting. In our experiments, we set the watermark sequence length $|u| = 40$.

**Details of consistency evaluation.** For perplexity, we compute the perplexity for each document in both the original IP dataset and the watermarked dataset. Moreover, we split each document into smaller blocks (50 words/block) and calculate the perplexity for each block individually. Extremely high perplexity indicates low-quality text or potential damage to the original content. We use these blocks to calculate the Filtering Rate, measuring the proportion of blocks filtered out when a perplexity threshold is applied. The QuRating employs a rating model to select high-quality data within the dataset, evaluating across four key dimensions: writing quality, facts&trivia, educational value, and required expertise.

### A.4 ADDITIONAL MAIN RESULTS

**Query one document multiple times.** In Table 1, we show that inserting only 12 canary documents into the IP dataset is sufficient to achieve strong detection performance when each canary document is queried only once. In this section, we want to explore whether inserting fewer canary documents into the IP dataset while querying a single canary document multiple times can still achieve effective detection. Specifically, we generate 14 different query questions for each canary document, allowing us to query a single canary document 14 times using diverse inputs. Table 8 presents the detection performance for different numbers of inserted canary documents, ranging from 1 to 5. The results show that even with just one canary document, we achieve 0.998 ROC-AUC and 96.6% TPR@1%FPR. By inserting only five canary documents, our method attains 1.000 ROC-AUC and 100% TPR@1%FPR, demonstrating the effectiveness of querying a single canary document multiple times to enhance detection performance.

Table 8: Detection performance for multiple queries per canary document multiple times. Specifically, we evaluate the detection performance by varying the number of inserted canary documents while querying each canary document 14 times.

| Canary Number | ROC-AUC | TPR@1%FPR | TPR@10%FPR |
| --- | --- | --- | --- |
| 1 | 0.998 | 0.966 | 0.998 |
| 2 | 0.999 | 0.988 | 1.000 |
| 3 | 0.999 | 0.996 | 1.000 |
| 4 | 0.999 | 0.998 | 1.000 |
| 5 | **1.000** | **1.000** | **1.000** |

**Robustness to canary paraphrasing attack.** We evaluate the robustness of our canaries by completely paraphrasing each one using GPT4o-mini and then measuring the detection performance using the paraphrased versions. As shown in the Table 9, even when all canary documents are paraphrased, our method still achieves strong detection performance with just 16 canaries. This result demonstrates the robustness and resilience of our method against modification.

**Robustness to a watermarked RA-LLM.** To further assess the robustness of our method, we consider a potential attack in which a malicious RA-LLM embeds its own watermarks into the outputs to erase the watermarking added to the canary documents. We consider the case where the RA-LLM applies the same watermarking technique with a different random key and evaluate the resulting de-

Table 9: Robustness of our method under the canary paraphrasing attack.

| Query Quota | ROC-AUC | TPR@1%FPR | TPR@10%FPR |
|---|---|---|---|
| 2 | 0.934 | 0.268 | 0.774 |
| 4 | 0.971 | 0.478 | 0.930 |
| 6 | 0.992 | 0.816 | 0.990 |
| 8 | 0.998 | 0.914 | 1.000 |
| 10 | 0.999 | 0.988 | 1.000 |
| 12 | 0.999 | 0.976 | 1.000 |
| 14 | 0.999 | 0.996 | 1.000 |
| 16 | 1.000 | 1.000 | 1.000 |

tection performance. The results are shown in the Table 10. As demonstrated, our method maintains strong detection performance even under this stronger adversarial setting.

Table 10: Robustness of our method against attacks where the RA-LLM employs the same watermarking technique with a different random key.

| Query Quota | ROC-AUC | TPR@1%FPR | TPR@10%FPR |
|---|---|---|---|
| 2 | 0.941 | 0.694 | 0.830 |
| 4 | 0.982 | 0.694 | 0.960 |
| 6 | 0.996 | 0.900 | 0.994 |
| 8 | 0.999 | 0.958 | 1.000 |
| 10 | 0.999 | 0.982 | 1.000 |
| 12 | 0.999 | 0.993 | 1.000 |
| 14 | 0.999 | 0.998 | 1.000 |
| 16 | 1.000 | 1.000 | 1.000 |

## A.5 ABLATION STUDY

**Ablation study on fictional and realistic canaries.** We conduct an ablation study to evaluate the effect of synthesizing fictional versus *realistic* canaries in our dataset synthesis pipeline on target retrieval accuracy (Figure 3). Specifically, we synthesize real entities (instead of fictional entities) and their description for realistic canaries. As shown in the Table 11, the realistic canaries cause the target retrieval accuracy to drop from 1.000 to 0.905, indicating a significant reduction in effectiveness. This result highlights the critical role of entity uniqueness in achieving reliable and accurate retrieval.

Table 11: Target retrieval accuracy for fictional and realistic canaries.

| Method | Original | WWJ | Ward | Ours-Fictional | Ours-Realistic |
|---|---|---|---|---|---|
| **Target Retrieval Accuracy** | 0.893 | 0.890 | 0.892 | 1.000 | 0.905 |

**Impact of watermark strength on detectability.** We examine the impact of different watermark strengths $\delta$ on detection performance. We set $\delta$ to 1, 2, and 3 and evaluate detection performance under the same query quota. As shown in Table 12, increasing $\delta$ enhances detection efficiency, allowing us to achieve 100% detection performance with fewer queries.

**Ablation study on a different RAG system.** To further demonstrate the universality and effectiveness of our method across different RAG systems, we evaluate it using an alternative RAG system, ANCE (Xiong et al., 2020a). The results are presented in the Table 13. Our method achieves consistently strong detection performance with only 10 canary documents, compared to 12 canary documents required when using Contriever-ms, highlighting the adaptability and universality of our approach across different retrievers.

**Detection performance with the larger base dataset.** We extended the base dataset size from $200,000$ to 2 million documents. As shown in the Table 14, our detection performance remains stable and unaffected despite the significant increase in dataset size. This result demonstrates the scalability and practical applicability of our method in real-world deployment scenarios.

**Detection performance when $K = 1$.** We conduct an ablation study on the selection parameter $K$ to further demonstrate the effectiveness of our method. In this study, we use a more challenging

Table 12: Detection performance of our method under different watermark strengths. As the watermark strength increases, our method achieves higher detection performance with fewer query quotas.

| Query Quota | $\delta = 1$ | | | $\delta = 2$ | | | $\delta = 3$ | | |
|---|---|---|---|---|---|---|---|---|---|
| | ROC-AUC | TPR@1%FPR | TPR@10%FPR | ROC-AUC | TPR@1%FPR | TPR@10%FPR | ROC-AUC | TPR@1%FPR | TPR@10%FPR |
| 1 | 0.855 | 0.153 | 0.556 | 0.910 | 0.294 | 0.724 | 0.962 | 0.616 | 0.894 |
| 2 | 0.937 | 0.276 | 0.786 | 0.970 | 0.546 | 0.908 | 0.993 | 0.86 | 0.986 |
| 4 | 0.981 | 0.610 | 0.948 | 0.995 | 0.850 | 0.996 | 0.999 | 0.986 | 1.000 |
| 6 | 0.995 | 0.808 | 0.998 | 0.998 | 0.960 | 1.000 | 0.999 | 0.994 | 1.000 |
| 8 | 0.999 | 0.962 | 1.000 | 0.999 | 0.994 | 1.000 | 1.000 | 1.000 | 1.000 |
| 10 | 0.999 | 0.992 | 1.000 | 0.999 | 0.998 | 1.000 | 1.000 | 1.000 | 1.000 |
| 12 | 0.999 | 0.998 | 1.000 | 1.000 | 1.000 | 1.000 | 1.000 | 1.000 | 1.000 |

Table 13: Detection performance of our method on the ANCE system.

| Query Quota | ROC-AUC | TPR@1%FPR | TPR@10%FPR |
|---|---|---|---|
| 1 | 0.907 | 0.312 | 0.732 |
| 2 | 0.975 | 0.604 | 0.944 |
| 4 | 0.994 | 0.872 | 0.990 |
| 6 | 0.998 | 0.988 | 1.000 |
| 8 | 0.999 | 0.996 | 1.000 |
| 10 | 1.000 | 1.000 | 1.000 |

setting by setting $K = 1$, meaning the RA-LLM retrieves only the single most relevant document from the dataset. Even under this stricter condition, our method achieves a retrieval accuracy of 0.995. The corresponding detection performance is shown in the Table 15. As we can see, even when RA-LLM is limited to retrieving only one document, our method continues to deliver strong detection performance with just 12 canary documents, which is consistent with the setting where $K = 3$.

### A.6 DETECTION PERFORMANCE WITH DIFFERENT WATERMARKING METHOD

We watermark the canary documents using KGW (Kirchenbauer et al., 2023)—the same technique used in Ward. As shown in the Table 16, our method still achieves 100% ROC-AUC with just 14 canaries, which is better than Ward. This further demonstrates the effectiveness of our proposed canary dataset synthesis algorithm.

## B THEORETICAL FALSE-ALARM GUARANTEE

**Theorem B.1** (False-alarm Guarantee). *[(Zhao et al., 2023, Theorem C.4)] Consider $\mathbf{y} = \mathbf{y}_{1:T}$ as any fixed suspect text. Let $N := |\mathcal{V}|$ and $G \subset \mathcal{V}$ satisfying $|G| = \gamma N$, where $\gamma \in [0, 1]$. $G$ is selected through the method described in Section 3.4, using a uniform random choice with a fixed key. Let $|\mathbf{y}|_G$ denote the number of tokens in $G$ and $z_{\mathbf{y}} = (|\mathbf{y}|_G - \gamma T)/\sqrt{\gamma(1-\gamma)T}$ as described in Section 3.5. Then the following statements hold:*

*1. Assume $T \geq 1$, then*

$$\mathbb{E}[|\mathbf{y}|_G \mid \mathbf{y}] = \gamma T \quad and \quad \mathbb{E}[z_{\mathbf{y}} \mid \mathbf{y}] = 0.$$

*2. Define $C_{\max}(\mathbf{y}) := \max_{i \in [N]} \sum_{j=1}^{T} \mathbf{1}(\mathbf{y}_j = i) \quad and \quad V(\mathbf{y}) := \frac{1}{T} \sum_{i=1}^{N} \left( \sum_{j=1}^{T} \mathbf{1}(\mathbf{y}_j = i) \right)^2$, then with probability $1 - \alpha$ (over only the randomness of $G$),*

$$\mathbb{P}\left[ |\mathbf{y}|_G \geq \gamma T + \sqrt{64\gamma T V \log(9/\alpha)} + 16C_{\max} \log(9/\alpha) \; \Big| \; \mathbf{y} \right] \leq \alpha,$$

*or equivalently (when $T \geq 1$),*

$$\mathbb{P}\left[ z_{\mathbf{y}} \geq \sqrt{\frac{64V \log(9/\alpha)}{1 - \gamma}} + \frac{16C_{\max} \log(9/\alpha)}{\sqrt{\gamma(1-\gamma)T}} \; \Big| \; \mathbf{y} \right] \leq \alpha.$$

Table 14: Detection performance with larger base dataset.

| Query Quota | ROC-AUC | TPR@1%FPR | TPR@10%FPR |
|---|---|---|---|
| 1 | 0.904 | 0.320 | 0.704 |
| 2 | 0.971 | 0.516 | 0.920 |
| 4 | 0.991 | 0.828 | 0.984 |
| 6 | 0.998 | 0.956 | 1.000 |
| 8 | 0.999 | 0.988 | 1.000 |
| 10 | 0.999 | 0.998 | 1.000 |
| 12 | 1.000 | 1.000 | 1.000 |

Table 15: Detection performance when $K = 1$.

| Query Quota | ROC-AUC | TPR@1%FPR | TPR@10%FPR |
|---|---|---|---|
| 1 | 0.888 | 0.252 | 0.662 |
| 2 | 0.954 | 0.428 | 0.846 |
| 4 | 0.989 | 0.748 | 0.980 |
| 6 | 0.997 | 0.900 | 1.000 |
| 8 | 0.999 | 0.986 | 1.000 |
| 10 | 0.999 | 0.998 | 1.000 |
| 12 | 1.000 | 1.000 | 1.000 |

With this theorem from Zhao et al. (2023), which upper bounds the false alarm probability, we can choose a test threshold $\eta > \sqrt{\frac{64V \log(9/\alpha)}{1-\gamma}} + \frac{16C_{\max} \log(9/\alpha)}{\sqrt{\gamma(1-\gamma)T}}$, then the false-alarm rate will be upper bounded by $\alpha$. Note that both $V$ and $C_{\max}$ can be computed from $\mathbf{y}$, allowing us to choose an input-dependent threshold to ensure a small enough False-alarm probability.

## C    PROMPTS USED IN CANARY DATASET SYNTHESIS ALGORITHM

---

### Attributes Extraction Prompt

### Task Description:
A reference text is given. You will carefully analyze the reference text and identify the following four key attributes.
1. Topic: Read the reference text and provide a high-level theme or general category of the reference text.
2. Subtopics: Based on the general topic, identify $\{n\}$ distinct general sub-category.
3. Writing Style: Analyze the overall writing style of the reference text.
4. Length Range: Provide an estimate of the length range of the reference text in terms of word count.

### Output Format Requirements:
Output the results with the JSON format (with four keys: topic, subtopics, writing_styles and length_range) and nothing else, such as {"topic": " ", "subtopics": [" ", " ", ...], "writing_styles": " ", "length_range": "m - n words"}.

### Reference Text:
$\{sampled\_text\}$

---

### Entity Synthesis

### Task Description:
1. Identify and list $\{n\}$ important entities mentioned within the reference text.
2. Synthesize $\{m\}$ fictional entities that align with the $\{sub\_topic\}$ topic.

### Synthesized Entities Requirements:
1. The synthesized entities should be creative and distinct.

---

Table 16: Detection performance with a different watermarking method.

| Query Quota | ROC-AUC | TPR@1%FPR | TPR@10%FPR |
|---|---|---|---|
| 1 | 0.833 | 0.362 | 0.619 |
| 2 | 0.920 | 0.588 | 0.760 |
| 4 | 0.980 | 0.744 | 0.948 |
| 6 | 0.994 | 0.938 | 0.982 |
| 8 | 0.998 | 0.962 | 0.994 |
| 10 | 0.999 | 0.986 | 0.998 |
| 12 | 0.999 | 0.998 | 0.998 |
| 14 | 1.000 | 1.000 | 1.000 |

2. Ensure the synthesized entities are fictional and do not overlap with real-world entities.

### Output Format Requirements:
Output the results with the JSON format (with two keys: real_entity and fictional_entity) and nothing else, such as {"real_entity": ["real_entity_1", "real_entity_2"] "fictional_entity": ["fictional_entity_1", "fictional_entity_2", ...]}.

### Reference Text:
$\{sampled\_text\}$

Description Synthesis

### Task Description:
1. Write $\{n\}$ fictional descriptions in an $\{writing\_style\}$ style about the following entities: $\{fictional\_entity\_1\}$, $\{fictional\_entity\_2\}$.
2. Create $\{m\}$ fictional interactions and discuss how those specified entities fictionally interact within the context of the $\{sub\_topic\}$ topic.

### Synthesized Description Requirements:
1. Create unique and imaginative content that has not been derived from existing material to avoid any issues with plagiarism.
2. Use creativity to simulate realistic scenarios that fit within the project's thematic boundaries.
3. Ensure factual accuracy where applicable, even in synthetic scenarios.
4. Incorporate diverse and inclusive content.
5. Do not mention "fictional" or any other indication that the entity or interaction is not real.

### Output Format Requirements:
Output the results with the JSON format, such as {"description_1": " ", "description_2": " "}.

Article Synthesis

### Task Description:
You are a content creator. You will be given some reference descriptions. You will carefully understand the reference descriptions and synthesize a text that satisfies the following instructions.
1. Generate a fictional text in the style of $\{writing\_styles\}$ in the context of $\{sub\_topic\}$ topic, with a length range of $\{length\_range\}$ in terms of word count.
2. Include the information in the given reference descriptions.

### Output Format Requirements:
Directly output the synthesized article in one paragraph and nothing else.

### Reference Descriptions:

{*reference_description*}

---

**Query Question Generation** Jovanović et al. (2024)

### Task Description:
Given an article, generate a question that can only be answered by reading the document. The answer should be a longer detailed response, so avoid factual and simple yes/no questions and steer more towards questions that ask for opinions or explanations of events or topics described in the documents. Do not provide the answer, provide just the question.

### Article:
{*canary_document*}

---

## D EXAMPLE OF CANARY DOCUMENT

Table 17: An example of a canary document, its corresponding query question, and the RA-LLM's response. The tokens highlighted in green indicate those belonging to the green list.

| | |
|---|---|
| Canary Document | The Flavonoid Research Institute (FRI) and the NutriQuest Study Group have collaborated on a landmark study to explore the relationship between flavonoid consumption and cardiovascular health, leveraging data from extensive epidemiological studies conducted by NutriQuest. By combining FRI's expertise in flavonoid analysis and bioactivity with NutriQuest's large-scale research designs, the partnership has provided a comprehensive understanding of how different flavonoid classes impact heart disease risk. The study's findings, which have been presented at an international conference, underscore the protective effects of flavonoids and emphasize the importance of public awareness regarding flavonoid-rich foods. Notably, this collaboration highlights the significance of interdisciplinary research, bringing together experts in epidemiology, biostatistics, and nutrition science to tackle the complex interplay between diet, lifestyle, and chronic diseases. The FRI's mission to promote awareness of flavonoid-rich foods aligns with NutriQuest's emphasis on inclusivity in research, ensuring that findings are representative and applicable to diverse demographic groups. Overall, this partnership has contributed significantly to the advancement of epidemiological research, ultimately informing evidence-based guidelines for dietary recommendations that can enhance public health outcomes. |
| Query Question | What key contributions can the collaboration between the Flavonoid Research Institute and the NutriQuest Study Group make to our understanding of cardiovascular health, and how might these findings influence public dietary recommendations? |
| Response | The collaboration between the Flavonoid Research Institute (FRI) and the NutriQuest Study Group has the potential to make significant contributions to our understanding of cardiovascular health, primarily by exploring the relationship between flavonoid consumption and heart disease risk. By combining FRI's expertise in flavonoid analysis and bioactivity with NutriQuest's large-scale research designs, the partnership can provide a comprehensive understanding of how different flavonoid classes impact heart disease risk. The study's findings, which have been presented at an international conference, underscore the protective effects of flavonoids and emphasize the importance of public awareness regarding flavonoid-rich foods. This collaboration highlights the significance of interdisciplinary research, bringing together experts in epidemiology, biostatistics, and nutrition science to tackle the complex interplay between diet, lifestyle, and chronic diseases. The findings of this collaboration can influence public dietary recommendations in several ways. Firstly, the study's results can inform evidence-based guidelines for dietary recommendations that can enhance public health outcomes. The FRI's mission to promote awareness of flavonoid-rich foods aligns with NutriQuest's emphasis on inclusivity in research, ensuring that findings are representative and applicable to diverse demographic groups. Secondly, the collaboration can raise public awareness about the importance of flavonoid-rich foods in reducing the risk of heart disease. By disseminating the critical role of flavonoids in mitigating heart disease risk, the partnership can encourage individuals to prioritize heart health by incorporating flavonoid-rich foods into their diets. Lastly, the study's findings can inform the development of nutritional supplements and products that cater to individuals at risk of cardiovascular diseases. The collaboration can also address inclusivity, ensuring accessibility across various demographics, including those with dietary restrictions. Overall, the collaboration between the Flavonoid Research Institute and the NutriQuest Study Group has the potential to make significant contributions to our understanding of cardiovascular health, and its findings can influence public dietary recommendations by promoting awareness of flavonoid-rich foods, informing evidence-based guidelines, and encouraging individuals to prioritize heart health. |

Table 18: An example of a canary dataset synthesis algorithm.

---

| | |
|---|---|
| Sampled Document | 2,4-Dinitrophenol (DNP) is reported to cause rapid loss of weight, but unfortunately is associated with an unacceptably high rate of significant adverse effects. DNP is sold mostly over the internet under a number of different names as a weight loss/slimming aid. It causes uncoupling of oxidative phosphorylation; the classic symptom complex associated with toxicity of phenol-based products such as DNP is a combination of hyperthermia, tachycardia, diaphoresis and tachypnoea, eventually leading to death. Fatalities related to exposure to DNP have been reported since the turn of the twentieth century. To date, there have been 62 published deaths in the medical literature attributed to DNP. In this review, we will describe the pattern and pathophysiology of DNP toxicity and summarise the previous fatalities associated with exposure to DNP. |
| Sub-topic | DNP Toxicity |
| Writing Style | Academic and Informative |
| Length Range | 150 - 200 words |
| Synthetic Entities | SlimSafe Elixir, MetaboliQ |
| Descriptions | Description 1: MetaboliQ is an innovative dietary supplement designed to enhance metabolic efficiency through a unique blend of botanical extracts and proprietary compounds. Its formulation includes ingredients known for their thermogenic properties, aiming to promote fat oxidation and energy expenditure. Clinical studies suggest that MetaboliQ may support weight management when combined with a balanced diet and regular exercise, although users are advised to approach its use with caution, particularly in light of the serious health risks associated with similar substances like 2,4-Dinitrophenol (DNP). |
| | Description 2: SlimSafe Elixir is marketed as a holistic weight loss solution, combining traditional herbal remedies with modern nutritional science. The elixir is formulated to assist in appetite regulation and promote a sense of satiety, utilizing a blend of adaptogenic herbs that are believed to balance hormonal responses related to hunger. While SlimSafe Elixir claims to offer a safer alternative to synthetic weight loss agents, it is essential for consumers to remain informed about the potential dangers of unregulated weight loss products, especially in the context of substances like DNP that have been linked to severe adverse effects. |
| | Description 3: In a recent study examining the interactions between various weight loss agents, researchers observed that the use of MetaboliQ alongside SlimSafe Elixir could potentially amplify the effects of both supplements. Participants reported increased energy levels and improved metabolic rates; however, the study also highlighted concerns regarding the cumulative impact of these products on cardiovascular health. Given the toxicological profile of DNP, it is crucial that individuals considering such combinations remain vigilant and consult healthcare professionals to mitigate risks associated with excessive stimulation of metabolic pathways. |
| Synthetic Article | The growing awareness of the dangers of 2,4-Dinitrophenol (DNP) has prompted a closer examination of dietary supplements that aim to enhance metabolic efficiency, such as MetaboliQ and SlimSafe Elixir. MetaboliQ, with its proprietary blend of botanical extracts, has been shown in clinical studies to support weight management when used in conjunction with a balanced diet and regular exercise, though caution is advised due to its potential thermogenic effects. SlimSafe Elixir, marketed as a holistic weight loss solution, utilizes adaptogenic herbs to regulate appetite and promote satiety, positioning itself as a safer alternative to synthetic agents like DNP, which have been linked to severe adverse effects. However, a recent study suggesting the potential for amplified effects when combining MetaboliQ with SlimSafe Elixir also highlights concerns regarding cumulative impacts on cardiovascular health, echoing the importance of vigilance and professional consultation to mitigate risks, particularly in light of DNP's toxicological profile. |
| Query Question | What are the potential risks and benefits of using MetaboliQ and SlimSafe Elixir for weight management, especially in light of the dangers associated with 2,4-Dinitrophenol (DNP), and how do these products compare to synthetic agents in terms of safety and effectiveness? |

