# OpenReview forum: "Dataset Protection via Watermarked Canaries in Retrieval-Augmented LLMs"
_ICLR.cc/2026/Conference — Submitted to ICLR 2026_

### Official Review · Reviewer_yvtf · 2025-10-30

**Soundness:** 3
**Presentation:** 3
**Contribution:** 3
**Rating:** 6
**Confidence:** 4

**Summary:**

The paper introduces CanaryTrace, a novel framework for protecting intellectual property datasets used in retrieval-augmented large language models. Instead of altering the original data, the method inserts synthetic, watermarked canary documents designed to be indistinguishable from the authentic dataset. These canaries are generated using a watermarked LLM and maintain consistency with the original dataset. When a potentially infringing RAG system is queried, the presence of the embedded watermark in model outputs serves as statistical evidence of dataset misuse. Empirically, CanaryTrace achieves 100% detection performance with only 12 queries, preserves dataset utility and maintains downstream RAG task performance across multiple datasets.

**Strengths:**

- The authors propose a novel canary strategy, which combines imperceptible watermarking and synthetic text generation, maintaining the original dataset completely intact.

- Extensive experiments across different datasets and retrievers demonstrate robustness, query efficiency, and minimal dataset distortion.

- The framework operates without access to model logits, making it deployable against closed-source RAG systems like GPT-5 or Gemini.

**Weaknesses:**

- Although the number of inserted canary documents is small, their presence could still introduce noise in large-scale retrieval systems. Normal user queries may inadvertently retrieve these synthetic canaries, slightly degrading retrieval precision or response quality in downstream applications.

- Despite efforts to ensure attribute-level consistency, there remains an inherent discrepancy between the original IP documents and the synthesized canaries. A sophisticated adversary could exploit subtle stylistic, semantic, or embedding-level differences to identify and filter out canary documents, thereby weakening the protection mechanism.

- The proposed z-test for watermark detection assumes that the queries used are independent. In practice, query correlation (e.g., overlapping semantics or shared retrieval results) could violate this assumption, leading to biased z-scores and inaccurate p-values, potentially affecting the reliability of the detection decision.

**Questions:**

See weaknesses.

---

> ### Author Response · Authors · 2025-11-21
>
> We sincerely thank you for your constructive comments. We are pleased that you found our canary strategy interesting and novel. We highly value your review and have carefully discussed your comments in the following responses.
>
> ### **Q1. The existence of canary documents may slightly degrade retrieval precision or response quality in downstream applications.**
>
> Thank you for pointing out this concern. We evaluate the impact of our method on downstream performance by measuring the correctness of model responses. Specifically, we use two datasets, DROP and Chinese-Poem, to represent two distinct task types. As shown in Table 4 of the manuscript, our method preserves the original dataset’s correctness without performance degradation, demonstrating that the inserted canaries do not affect the downstream quality of RAG outputs.
>
> ### **Q2. A sophisticated adversary could identify and filter out canary documents, thereby weakening the protection mechanism.**
>
> Thank you for pointing out this concern. There is indeed a cat-and-mouse dynamic: an evolving game between defenders and adversaries. Currently, we have introduced the canary pre-filtering method (detailed in Appendix A.2) to screen out suspicious canaries and improve the resistance of the remaining canaries to different kinds of attacks.
>
> Moreover, we would like to highlight that, while a detailed filtering can be a potentially effective method for detecting canary documents, it may require significant computational resources, cost, or human effort, especially when applied to large-scale datasets containing more than a million documents and only dozens of canaries. In contrast, the cost of synthesizing and inserting canary documents is negligible. This raises an important consideration: in real-world scenarios, it is questionable whether an adversary would be willing to invest substantial effort and resources solely to identify and remove a small number of canaries. If the cost of removal exceeds that of simply purchasing a license for the dataset, our approach remains a cost-efficient and practical solution for protecting IP datasets.
>
> ### **Q3. Query correlation may violate the independent assumption of the z-test, leading to biased z-scores and inaccurate p-values, potentially affecting the reliability of the detection decision.**
>
> We thank the reviewer for this insightful comment.  We agree that the query correlation could violate the independence assumption, potentially inflating z-scores. However, our method is explicitly designed to satisfy the independence assumption through the construction of the canary dataset. **One of our key objectives is to minimize the semantic overlap between individual data points.** To ensure that, our design of canary data includes both fictitious entities and relations. Moreover, **the independence of the queries is empirically supported by our Target Retrieval Accuracy results (Figure 3 in our manuscript)**, in which our method achieves 100% retrieval accuracy. Since the retrieved contexts (the canaries) are distinct and non-overlapping, the resulting responses are generated based on different source texts. Therefore, the watermark signals in the concatenated responses remain statistically independent, validating the use of the z-test.
>
> Moreover, we empirically validate the detection performance in a situation that one canary is queried many times (Table 8 in Appendix A.4). Specifically, we generate 14 different queries for one canary, and query one canary 14 times with those different queries. The results demonstrate that by inserting only five canary documents, our method achieves 1.000 ROC-AUC and 100% TPR@1%FPR, demonstrating the effectiveness of our method by querying a single canary document multiple times.

---

### Official Review · Reviewer_MqVp · 2025-10-31

**Soundness:** 3
**Presentation:** 3
**Contribution:** 3
**Rating:** 6
**Confidence:** 4

**Summary:**

The paper introduces CanaryTrace, a dataset-ownership protection scheme for RAG systems. It inserts a small set of synthetic, watermarked canary documents that (1) match the attributes of the dataset, (2) are fictional to ensure the uniqueness for high retrievability, and (3) are robust enough so their content causes a detectable watermark signal to diffuse into RAG responses when retrieved. Experiments show strong target-retrieval accuracy and detection, while leaving downstream RAG accuracy intact.

**Strengths:**

1.The DMI-RAG task is well-posed, which keeps the original dataset untouched, avoids quality regressions common to paraphrase-based watermarking, and canaries provide separable evidence

2.The work provides a principled synthesis pipeline.

3.Detection with statistical guarantees.

4.Extensive experiments and strong empirical results under realistic constraints.

**Weaknesses:**

1. More broader evaluation of robustness should be done. How does detection performance change if the generator performs an aggressive attack, e.g., paraphrasing, before answering?

2. The experiments show the effectiveness of the watermarked canary on text RAG. Can you discuss its potential application to non-text RAG, e.g., image?

**Questions:**

Have you considered a standardized disclosure (e.g., a notarized canary list and key escrow) to bolster evidentiary value?

How should practitioners choose $\eta$ to achieve, say, 1% FPR?

---

> ### Author Response · Authors · 2025-11-21
>
> We sincerely thank you for the detailed feedback. We are pleased that you recognize the soundness of our task formulation, the principled synthesis pipeline, the statistically grounded detection framework, and the strength of our experimental results under realistic settings. We highly value your review and have carefully discussed your comments in the following responses.
>
>
> ### **Q1. More broader evaluation of robustness should be done. How does detection performance change if the generator performs an aggressive attack?**
>
> We evaluate the robustness of our canaries by completely paraphrasing each one using GPT4o-mini and then measuring the detection performance using the paraphrased versions. As shown in the following table, even when all canary documents are paraphrased, our method still achieves strong detection performance with just 16 canaries. This result demonstrates the robustness and resilience of our method against modification.
>
> | Query Quota | ROC-AUC | TPR@1%FPR | TPR@10%FPR |
> |:-----------:|:-------:|:---------:|:----------:|
> |      2      |  0.934  |   0.268   |    0.774   |
> |      4      |  0.971  |   0.478   |    0.930   |
> |      6      |  0.992  |   0.816   |    0.990   |
> |      8      |  0.998  |   0.914   |    1.000   |
> |      10     |  0.999  |   0.988   |    1.000   |
> |      12     |  0.999  |   0.976   |    1.000   |
> |      14     |  0.999  |   0.996   |    1.000   |
> |      16     |  1.000  |   1.000   |    1.000   |
>
>
> To further assess the robustness of our method, we consider a potential attack in which a malicious RA-LLM embeds its own watermarks into the outputs to erase the watermarking added to the canary documents. We consider the case where the RA-LLM applies the same watermarking technique with a different random key and evaluate the resulting detection performance. The results are shown in the following table. As demonstrated, our method maintains strong detection performance even under this stronger adversarial setting.
>
> | Query Quota | ROC-AUC | TPR@1%FPR | TPR@10%FPR |
> |:-----------:|:-------:|:---------:|:----------:|
> |      2      |  0.941  |   0.694   |    0.830   |
> |      4      |  0.982  |   0.694   |    0.960   |
> |      6      |  0.996  |   0.900   |    0.994   |
> |      8      |  0.999  |   0.958   |    1.000   |
> |      10     |  0.999  |   0.982   |    1.000   |
> |      12     |  0.999  |   0.993   |    1.000   |
> |      14     |  0.999  |   0.998   |    1.000   |
> |      16     |  1.000  |   1.000   |    1.000   |
>
>
> ### **Q2. The experiments show the effectiveness of the watermarked canary on text RAG. Can you discuss its potential application to non-text RAG, e.g., image?**
>
> Thank you for your good suggestion. Extending our method to image RAG is a promising direction, with two key challenges. (1) Canary synthesis: we must generate canary images that achieve high retrieval accuracy while remaining distributionally consistent with the target corpus. (2) Watermark robustness: the watermark embedded in the canary must remain detectable end-to-end, i.e., it should survive retrieval, context assembly, and generation so that the watermark signal is recoverable from the model’s final output. We will explore this direction in future work.
>
> ### **Q3. Have you considered a standardized disclosure (e.g., a notarized canary list and key escrow) to bolster evidentiary value?**
>
> We agree that standardized disclosure can strengthen the evidentiary value of our approach. Our method already provides a keyed statistical signal and a formal false-alarm control over the responses, which makes the reproducible third-party verification easy. We believe the standardized disclosure will increase the practicality of our method in real-world applications.
>
> ### **Q4. How should practitioners choose $\eta$ to achieve, say, 1% FPR?**
>
> Our Theorem B.1 provides exactly this: it gives a direct mapping from a desired FPR to the decision threshold $\eta$. In practice, a practitioner can follow three steps: (1) Choose the target FPR (e.g., 1%); (2) Set the $\eta$ using Theorem B.1. (3) Optionally run a small calibration on known unwatermarked texts to refine the threshold for more accurate control. This procedure keeps FPR at or below the desired FPR.

---

### Official Review · Reviewer_KykN · 2025-10-31

**Soundness:** 3
**Presentation:** 3
**Contribution:** 2
**Rating:** 4
**Confidence:** 3

**Summary:**

This paper addresses dataset copyright protection in Retrieval-Augmented Generation systems, which risk unauthorized incorporation of intellectual property datasets into Retrieval-Augmented LLMs. The authors propose CanaryTrace, a novel method for detecting dataset misuse. CanaryTrace works by inserting synthetic documents that contain unique, statistically provable embedded watermark into the original dataset without altering it. When unauthorized use occurs, querying these canaries from the RA-LLM reveals statistical evidence of the watermark. Experiments show high query efficiency, detectability, and consistency, with minimal impact on dataset quality or RAG performance.

**Strengths:**

- The paper introduces a black-box dataset protection framework that preserves the integrity of the original IP dataset while achieving high detection accuracy through LLM-based watermarking in synthetic canary documents

- The authors conduct extensive experiments demonstrating strong quantitative performance, including high retrieval accuracy with minimal queries and negligible impact on downstream RAG tasks

**Weaknesses:**

- The proposed method lacks clear novelty. Its core idea primarily relies on applying existing watermarking techniques within the RAG framework. While the implementation is well-executed, the approach essentially extends known watermarking methods to a familiar setting without introducing fundamentally new algorithms or theoretical insights.

- The method used to detect the watermark also lacks clear novelty. This paper primarily applies an existing detection algorithm to detect the watermark.

**Questions:**

What is the retriever model used for the experiments?

---

> ### Author Response · Authors · 2025-11-21
>
> We sincerely thank you for your constructive feedback. We highly value your review and provide our detailed responses below.
>
> ### **Q1&Q2. Clarify the contribution and novelty of the proposed method.**
>
> **We believe the reviewer may overlook the core contribution of our work, and we would like to take this opportunity to clarify the key innovations introduced in our approach.** Specifically, our key contribution is not a new watermark technique, instead a new dataset protection method for the RAG setting. Concretely,
>
> - We formulate the DMI-RAG problem and cast detection as black-box hypothesis testing over concatenated RAG responses.
>
> - Most importantly, we propose a **new canary dataset synthesis method** to synthesize new data by extracting key attributes from the original IP data. To the best of our knowledge, **our work is the first to propose using a synthesized and watermarked canary dataset to protect intellectual property from unauthorized use by RA-LLMs**, which has not been explored in prior work.
>
> - Empirically, our method **ourperforms baselines** and achieves: (1) a significant **improvement in retrieval accuracy** compared with baselines (from 89.2% to 100%; see Figure 3 in manuscript), (2) **stronger detection performance** using fewer queries compared with baselines (Table 1 in manuscript), (3) **low dataset distortion** (Table 2), and (4) **high consistency** (Table 3 and Figure 4).
>
> We thank the reviewer for pointing out this concern. We will further revise our manuscript to make our contribution clearer.
>
>
> ### **Q3. What is the retriever model used for the experiments?**
>
> Thank you for pointing out this problem. For retriever, we conduct experiments based on the Contriever-ms and ANCE, which has been specified in Section 4.1.

---

### Official Review · Reviewer_djSQ · 2025-11-04

**Soundness:** 2
**Presentation:** 3
**Contribution:** 2
**Rating:** 2
**Confidence:** 4

**Summary:**

This paper proposes a novel method for dataset copyright protection in retrieval-augmented large language models (RAG-LLMs) via the use of watermarked canaries. The key idea is to embed special, inconspicuous textual fragments (canaries) into the retrieved data with imperceptible yet verifiable watermarks. These canaries enable the dataset owner to later verify unauthorized usage or leakage of their dataset in an LLM system, even when the model or retriever is proprietary.

The authors design semantic-preserving watermarking techniques to inject canaries without degrading retrieval or generation quality.

They introduce an adaptive embedding mechanism that selects insertion points based on retrieval relevance and semantic alignment.

A verification framework detects the presence of watermarked canaries in LLM outputs through probabilistic decoding analysis and semantic matching.

The approach is evaluated on multiple RAG settings (e.g., GPT-4-retrieval hybrid, open-source retrievers).

**Strengths:**

1. address a timely problem for protecting the intellictual property for external knowledge base used for the RAG system.

2. The results evaluated on various dataset is really good.

3. The presentation is easy to understand.

**Weaknesses:**

1. There is no thereotical guarantee that this unique  (e.g., "VitalityBoost and ExerciseShield " ) will trigger the retrieval of systhetic paragraph. For example, what happens if these unique content (e.g., "VitalityBoost and ExerciseShield " ) will exist in other  external knowledge bases which would be combined into the protected dataset.

2. How do you verify the ownership with answers containinng the sythentic content. It is possible that the user claim that their knowledge base contains these contents? Different from other backdoor-based defense, they have malicious behaviour used as verification evidence.

3. I here challenge the novely of this paper as it seems a variant of previous work [1]. Both of these work use generated / systhetic paragraph / content as evidences for ownership. I would like to see the merits or disadvantages of this work comparing with previous work.

4. There is no evaluation on real-world IP-sensitive database such as Harry Potter series book.

5. Lack important baselines, such as membership inference-based approaches and a general framework for data use auditing (CCS).

6. How can you ensure your question will accurately retrieve the synthetic content aas you expected with varying k and the external datasets ?


[1] Towards copyright protection for knowledge bases of retrieval-augmented language
models via ownership verification with reasoning. arXiv preprint arXiv:2502.10440, 2025

**Questions:**

See Above.

---

> ### Author Response · Authors · 2025-11-21
>
> We sincerely thank you for your constructive comments. We highly value your review and provide our detailed responses below.
>
> ### **Q1&Q6. How to ensure the uniqueness of the canary? How can you ensure your question will accurately retrieve the canary as you expected with varying k and the external datasets?**
>
> Thank you for raising this important concern. We provide the following clarification to address your concern.
>
> - Black-box setting: **Our method does not require every query to be accurate in this black-box RAG setting; instead, our guarantees are based on statistical watermarking.** We **aggregate evidence across multiple queries** via the z-statistics by detecting the watermarking, ensuring that occasional retrieval misses do not affect the overall detection reliability. Moreover, our statistical detection method provides a strict false alarm control guarantee, as shown in Appendix B.
>
> - Compound uniqueness: Crucially, we would like to clarify that our canaries are designed for **compound uniqueness**, including **not only multiple fictitious entities but also their interrelations.** Moreover, **our specifically designed queries target both those surface entities and their relational context, which substantially reduces the chance that an external knowledge base containing the same entity names will be retrieved by mistake.**
>
> - Strong empirical evidence: Empirically, we demonstrate that our method achieves **100% target retrieval accuracy** (Figure 3 in manuscript) and 100% FPR at 1% FPR with only 12 queries (Table 1 in manuscript) on the NFCorpus dataset, indicating that the collisions are rare in practice. Moreover, **our results still show strong performance under the stricter $K=1$ retrieval regime**, where $K$ denotes the number of top documents retrieved for each query. It achieves a **retrieval accuracy of 99.5%** and 100% FPR at 1% FPR with only 12 queries (Table 15 in manuscript).
>
> Overall, the combination of compound uniqueness, relation-focused queries, and statistical aggregation yields strong retrieval and detection in realistic mixed-corpus settings, which extremely reduces the semantic overlap and has been demonstrated empirically.
>
> ### **Q2. How do you verify the ownership with answers containing the synthetic content? Is it possible that the user claims that their knowledge base contains these contents?**
>
> Thank you for pointing out this concern. **It seems there may be some misunderstanding of our method.** Importantly, we would like to clarify that **our detection does not rely on the presence of specific synthetic content in the model response.** Instead, the ownership is established through the **detection of the imperceptible, statistically verifiable watermark** embedded within the synthetic canaries and later observed in the model’s response. The watermark pattern is generated using a private key and verified via a formal hypothesis test with **theoretical false-alarm control** (Appendix B), providing objective and reproducible statistical evidence of unauthorized dataset usage. Therefore, even if a third-party corpus independently contained similar fictional names, the probability that its responses would also reproduce the correct watermark pattern is exponentially small.

---

> ### Author Response · Authors · 2025-11-21
>
> ### **Q3. Comparison with [1]**
>
> We sincerely thank the reviewer for providing a concurrent and related work, and we carefully read the provided paper. **We respectfully note that the reviewer may have some misunderstanding of our method.** Specifically, the reviewer claims that ‘Both of these works use generated/synthetic paragraph/content as evidence for ownership’. However, we would like to clarify that **our method does not rely on the presence of content in the model response as evidence for ownership.** Instead, the ownership is established through the **detection of the imperceptible, statistically verifiable watermark** embedded within the synthetic canaries and later observed in the model’s response. **This is completely different from the provided work [1].**
>
> Moreover, even though both works study the dataset protection, they use completely different methods, especially in the following aspects.
>
> (1) Evidence carrier:
> - [1]: Uses Chain-of-Thought (CoT); verifies ownership by checking whether responses reproduce the target CoT.
> - Ours: Uses a keyed token-level watermark that appears in the final answers when watermarked canary paragraphs are retrieved.
>
> (2) Test & guarantees:
> - [1]: Judge presence/absence of the target CoT content.
> - Ours: Perform a z-test on concatenated outputs with a theoretical false-alarm guarantee.
>
> (3) Data treatment:
> - [1]: Adds CoT templates/phrases tied to specific queries into the dataset.
> - Ours: Inserts a small set of synthetic canary documents (fictional entities + relations) designed for reliable retrieval.
>
> In addition to the provided work [1], we also notice four more related works [2-5]. **We ensure we will include all those related works and corresponding discussions in our new manuscript.**
>
> [1] ​​Guo, Junfeng, Yiming Li, Ruibo Chen, Yihan Wu, Chenxi Liu, Yanshuo Chen, and Heng Huang. "Towards copyright protection for knowledge bases of retrieval-augmented language models via ownership verification with reasoning." arXiv preprint arXiv:2502.10440 (2025).
>
> [2] Li, Yiming, Shuo Shao, Yu He, Junfeng Guo, Tianwei Zhang, Zhan Qin, Pin-Yu Chen et al. "Rethinking data protection in the (generative) artificial intelligence era." arXiv preprint arXiv:2507.03034 (2025).
>
>
> [3] Guo, Junfeng, Yiming Li, Ruibo Chen, Yihan Wu, and Heng Huang. "Zeromark: Towards dataset ownership verification without disclosing watermark." Advances in Neural Information Processing Systems 37 (2024): 120468-120500.
>
> [4] Golatkar, Aditya, Alessandro Achille, Luca Zancato, Yu-Xiang Wang, Ashwin Swaminathan, and Stefano Soatto. "Cpr: Retrieval augmented generation for copyright protection." In Proceedings of the IEEE/CVF conference on computer vision and pattern recognition, pp. 12374-12384. 2024.
>
> [5] Qi, Zhenting, Hanlin Zhang, Eric Xing, Sham Kakade, and Himabindu Lakkaraju. "Follow my instruction and spill the beans: Scalable data extraction from retrieval-augmented generation systems." arXiv preprint arXiv:2402.17840 (2024).
>
> ### **Q4. Evaluation on a real-world IP-sensitive database.**
>
> Thank you for your good suggestion. To further assess the detection performance of our method on real-world IP-sensitive data, we conducted an additional experiment using such a dataset.
>
> Similar to [1], we select the Wikipedia dataset [2] as the copyright-involved dataset to evaluate the detection performance of our method. It demonstrates that our method consistently achieves high detection performance on the copyright-involved dataset.
>
> | Query Quota | ROC-AUC | TPR@1%FPR | TPR@10%FPR |
> |:-----------:|:-------:|:---------:|:----------:|
> |      2      |  0.971  |   0.535   |    0.910   |
> |      4      |  0.995  |   0.863   |    0.995   |
> |      6      |  0.998  |   0.962   |    1.000   |
> |      8      |  0.999  |   0.997   |    1.000   |
> |      10     |  0.999  |   0.999   |    1.000   |
> |      12     |  1.000  |   1.000   |    1.000   |
>
>
>
> [1] ​​Guo, Junfeng, Yiming Li, Ruibo Chen, Yihan Wu, Chenxi Liu, Yanshuo Chen, and Heng Huang. "Towards copyright protection for knowledge bases of retrieval-augmented language models via ownership verification with reasoning." arXiv preprint arXiv:2502.10440 (2025).
>
> [2] Weber, Maurice, Dan Fu, Quentin Anthony, Yonatan Oren, Shane Adams, Anton Alexandrov, Xiaozhong Lyu et al. "Redpajama: an open dataset for training large language models." Advances in neural information processing systems 37 (2024): 116462-116492.

---

> ### Author Response · Authors · 2025-11-21
>
> ### **Q5. Reason for not using membership inference-based approaches and [3] as baselines**
>
> Thank you for pointing out this concern. Our approach is an **active protection mechanism**: before release, we embed imperceptible, keyed watermarks into a small set of synthetic canary documents that leave the original IP data untouched; later, we verify ownership by detecting the watermark information in the final answers of a black-box RA-LLM. In contrast, the membership-inference approaches are passive: they attempt to infer usage without any pre-embedded signal. Therefore, we didn’t include the membership-inference-based methods as baselines because it is an unfair comparison given the different problem setting. **Our current baselines, by contrast, all fall under the category of active protection methods**, which proactively embed identifiable signals into the dataset prior to release. Moreover, the method proposed in [3] focuses on auditing data use during machine-learning model training, a setting fundamentally distinct from our RAG context.
>
> We thank you for your valuable suggestion. We will further revise our manuscript to highlight this difference, and we will discuss [3] in our manuscript.
>
> [1] Li, Yuying, Gaoyang Liu, Yang Yang, and Chen Wang. "Seeing is believing: Black-box membership inference attacks against retrieval augmented generation." arXiv e-prints (2024): arXiv-2406.
>
> [2] Anderson, Maya, Guy Amit, and Abigail Goldsteen. "Is my data in your retrieval database? membership inference attacks against retrieval augmented generation." arXiv preprint arXiv:2405.20446 (2024).
>
> [3] Huang, Zonghao, Neil Zhenqiang Gong, and Michael K. Reiter. "A general framework for data-use auditing of ML models." In Proceedings of the 2024 on ACM SIGSAC Conference on Computer and Communications Security, pp. 1300-1314. 2024.

---

### Meta-Review · Area_Chair_ubxn · 2026-01-03

**Summary:**

This paper presents CanaryTrace, a method for protecting RAG datasets by inserting watermarked, attribute-mimicking synthetic canary content, with a clear motivation and substantial experimental validation. The authors have adequately addressed the majority of issues and weaknesses raised by reviewers. However, as noted by Reviewer djSQ and Reviewer KykN, there remain gaps in the comprehensive comparison with prior work and clear novelty.

**Reviewer Concerns:**

## Addressed Concerns:

### Reviewer djSQ

- Methodological details including thereotical guarantee, ownership verification and Retrieval stability: Resolved. The explanation has been largely clarified.
- Validated on real-world IP sensitive datasets. Resolved. The author added the "real-world IP-sensitive database" experiment.
- The differences from membership inference-based approaches: Resolved. The authors added relevant statements.

### Reviewer KykN

- The retriever model used for the experiments: Resolved. The authors provided the model.

### Reviewer MqVp

- Robustness assessment needs to be expanded: Resolved. The authors have provided additional experiments to evaluate the robustness.
- Questions regarding application scenario expansion: Resolved. The author mentioned some scenarios and plans to consider them in future work.
- Lack of standardized disclosure and parameter selection guidance: Resolved. The authors added relevant statements.

### Reviewer yvtf

- Sophisticated adversary risks: Resolved. The authors added relevant statements.
- Detection of assumption limitations: Resolved. The authors added relevant statements.

## Outstanding Concerns:
### Reviewer djSQ

- The selection of the baseline and a comprehensive comparison with prior work: Not Resolved. No experimental analysis was conducted on the method mentioned in Reviewer Comment Q3 to highlight its innovation (rather than being a variant of previous work).

### Reviewer KykN

- The watermark lacks clear novelty and the paper directly applies an existing detection algorithm to detect the watermark: Not Resolved. The author did not reply the issue of directly adopting prior watermark detection methods.

### Reviewer yvtf
- Retrieval precision and response quality: Not Resolved. Did not directly address whether false positives exist.

**Reviewer Scores:**

I think if the reviewer djSQ actively participates in the discussion, his score will improve. And others may maintain their original scores.

---

### Decision · Program_Chairs · 2026-01-26

Reject